# Level of Physical Activity in Pregnant Populations from Different Geographic Regions: A Systematic Review

**DOI:** 10.3390/jcm11154638

**Published:** 2022-08-08

**Authors:** Cristina Silva-Jose, Miguel Sánchez-Polán, Rubén Barakat, Javier Gil-Ares, Ignacio Refoyo

**Affiliations:** 1AFIPE Research Group, Faculty of Physical Activity and Sport Sciences-INEF, Universidad Politécnica de Madrid, 28040 Madrid, Spain; 2Sports Department, Faculty of Physical Activity and Sports Sciences-INEF, Universidad Politécnica de Madrid, 28040 Madrid, Spain

**Keywords:** pregnancy, physical activity, health promotion, active, prevalence

## Abstract

The aim of this study was to examine the level of physical activity during pregnancy in different populations worldwide. An intensive search was carried out from February until May 2021. The inclusion criteria were original studies of healthy pregnant women, and the main study variable was the assessment of physical activity. A total of 110 out of 1451 studies were assessed for inclusion, using the Newcastle–Ottawa Scale for quality, and for the risk of bias. The 44 analyzed articles were divided into 5 tables according to the characteristics of the intervention and the validated instrument used to measure physical activity (PA). A total of 59.09% of the studies indicated that participants had a low level of physical activity during pregnancy. In addition, the median quality score of the studies was 7.12, and 77.27% of the studies were cataloged as having a high-quality score. Although international guidelines recommend that women without a contraindication engage in prenatal physical activity, the results of the present study show that the level of PA is too low for women to achieve scientifically proven maternal-fetal benefits. Failure to achieve the recommended levels of weekly physical activity could pose significant risks to maternal well-being.

## 1. Introduction

Today’s society shows concerning and increasing levels of physical inactivity, which are associated with an alarming number of complications and associated pathologies, such as gestational diabetes or preeclampsia [1,2]. In parallel, people are experiencing higher levels of stress in the body and more health problems year after year, which support the appearance of diseases that reduce the quality of life [3,4] as well as high globalized economic costs; these diseases are directly responsible for the high cost of medicines, hospital stays and clinical consultations [5].

The gestational period is not excluded from this problem [6], which supports the appearance of diseases. A relevant consequence is the saturation of the health system with complications and pathologies that could be addressed with healthy lifestyle habits, good nutrition, and the regular practice of physical activity [7].

It should be added that pregnancy is considered a period with great influence on the establishment of certain healthy habits that the woman will continue (in some cases throughout her life) [8]; therefore, comprehensive wellness interventions are necessary. Balance in pregnant women’s systems is important to avoid maternal, fetal and newborn risks and ensure a healthy pregnancy. In this sense, physical inactivity can be a threat to this desired and complex situation [9], and therefore, it would be interesting to know the level of compliance to physical activity guidelines affecting the health of the mother and fetus [10]. Pregnant women spent more than 50% of their time in sedentary behaviors and it impacted on pregnancy outcomes for both mother (higher levels of C Reactive Protein or LDL Cholesterol) and child (larger newborn abdominal circumference or macrosomic infants) [11].

Increasingly, in recent years, the World Health Organization has been working on the creation of guidelines and recommendations for the practice of physical activity in the fight against high levels of sedentarism [12], and specifically, there are global organizations that, for more than 30 years, have been dealing with the complex relationship between physical activity and pregnancy for healthy pregnant women, publishing international clinical guidelines for physical exercise and pregnancy [13,14]. Specifically, recent universal guidelines (American, Canadian, Spanish), in consensus with others throughout the world, have established the recommendation for a healthy pregnant woman to stay physically active, performing 150 min of moderate physical activity at least 3 days a week and, whenever possible, under the supervision of a professional [13,14,15]. At this point, the current scientific consensus emerging from early evidence [16,17] suggests that an accumulation of 30 min or more of moderate exercise per day should occur on most, if not all, days of the week in the absence of either medical or obstetric complications, similar to the recommendation for the non-pregnant population.

In a global analysis, the scientific literature has confirmed a positive association between the regular practice of moderate physical activity by healthy pregnant women and maternal, fetal and newborn well-being [18]. Similarly, and in direct relation to the health of the mother and her future child, scientific evidence speculates that an active pregnancy produces an epigenetic effect of physical activity as a promoter of a healthy and balanced intrauterine environment as the basis of health, both during the gestational period and after it [19,20]. Accordingly, the objective of the present study was to examine the level of physical activity of the pregnant population in different regions of the world through the analysis of the scientific literature on this issue to be aware of the basis for future interventions.

## 2. Materials and Methods

This was a systematic review designed based on the Preferred Reporting Items for Systematic Reviews and Meta-Analyses guidelines (P.R.I.S.M.A.) [21]. The protocol was registered in the International Prospective Register of Systematic Reviews (PROSPERO) (CRD42021262193).

### 2.1. Data Sources

An intensive search was carried out in the following databases: PubMed, Scopus, Sport Discus and Web of Science. The search began in February 2021 and ended in April 2021. To guarantee equality in the selection process, the same article selection criteria were used for all databases considering the differences in controlled vocabulary and syntax rules. The detailed search strategy is presented in Appendix A.

The reference lists of the selected studies were reviewed to identify other studies that may have been missed in the electronic keyword search. The registries’ observational studies were reviewed to identify unpublished research. The search terms used were:(“physical activity” OR “exercise” OR “training”)(“barriers” OR “enablers” OR “access”)(“pregnancy” OR “maternal” OR “antenatal” OR “pregnant”)(“health” OR “wellbeing”)(“maternal outcome” OR “pregnancy outcome”)(“observational” OR “cohort” OR “qualitative” NOT “randomized clinical trial” NOT “RCT”)

### 2.2. Study Inclusion and Exclusion Criteria

To address the main objective, study inclusion and exclusion criteria were structured using the PICOS framework (population, intervention, comparison, outcome, study design) as a worksheet [22] (Table 1).

Population:

The population of interest was healthy pregnant women without contraindications to physical exercise, regardless of their gestational age at the time of entry to the study. The characteristics of the population were reviewed in the methodology of each study. Women with contraindications were excluded from the analyses [13,14,15].

Intervention:

The investigated intervention was related to the different forms of assessment, analysis and recording of the prevalence of physical activity during pregnancy in different regions of the world. Due to the different nature of the studies and, therefore, their different forms of interventions, studies whose measurement of physical activity was assessed through validated questionnaires, including the Pregnancy Physical Activity Questionnaire (PPAQ) [23], the International Physical Activity Questionnaire (IPAQ) [24], and the Kaiser Modified Questionnaire (KPAS) [25], or by direct and mixed measurements (accelerometer and pedometer) [26,27] were included (Table 2).

Comparison:

In this case, the comparison was based on not practicing physical activity. For this comparison, women who engaged in physical activity were compared to those who did not.

Outcome:

The primary outcome was the level of physical activity performed during pregnancy, and the secondary outcomes were variables corresponding to the mother and the newborn (Table 3, Table 4, Table 5, Table 6 and Table 7). The studies must have contained at least one primary study variable that somehow recorded the quantity of physical activity performed during pregnancy using the aforementioned validated questionnaires or direct measurement methods.

Study selection:

In all cases, observational studies were selected, and studies related to interventions (randomized clinical trials or quasi-randomized clinical trials), some type of review (narrative, systematic or systematic review with meta-analysis) and qualitative research were excluded. In addition, articles published between 2006 and 2021 and written in English, Spanish and Portuguese were considered for the search.

### 2.3. Data Extraction

The selection process that was followed for the reviewed articles is captured in Figure 1 [28]. Titles and abstracts identified in the electronic searches were independently screened by two researchers to select potentially relevant studies. The abstracts were identified and passed an initial analysis, and posteriori searches of the full text were performed. The full texts were analyzed separately to search for priority outcomes for data extraction. Reference lists of selected articles were checked to identify possible additional studies not identified by electronic searches.

For studies where one of the investigators recommended their exclusion, both investigators agreed by consensus to make a final decision on whether they were included. In situations of absolute discrepancy, a third researcher provided his assessment on the inclusion of the study.

Finally, 110 studies from a total of 1451 studies were assessed for inclusion by two independent reviewers. Quality assessment and risk of bias were assessed following the Newcastle–Ottawa Scale (NOS), and studies classified as high or moderate quality were included [29,30]. This scale establishes an overall quality score that ranges from 0 to 9 stars using three factors in its measurement: selection, comparability, and results. A “high” quality score required 3 or 4 stars for selection, 1 or 2 stars for comparability, and 2 or 3 stars for results, and a “moderate” quality score required 2 stars for selection, 1 or 2 stars for comparability, and 2 or 3 stars for results [31]. Studies with lower scores were discarded.

### 2.4. Data Synthesis

Given our goal of knowing the levels of physical activity of the pregnant population in the existing literature, the heterogeneity in the study designs and in the techniques assessed and the measurement tools for the main variable, a narrative synthesis was carried out precluding the conduct of a formal metanalysis. The 44 analyzed articles were divided into 5 tables according to the characteristics of the intervention and the instrument used to measure physical activity.

As shown in Table 3, Table 4, Table 5, Table 6 and Table 7, the data extracted from each study were divided into three groups. First, the data on the author(s), type of study, sample size, year of publication and country where the study was carried out were used to determine the methodological design. Then, data from the study group, such as gestational period, and maternal outcomes were used to define the characteristics of the sample. Finally, the physical activity measuring instrument, definition of physical activity and main conclusions were used to determine the method of measuring the primary outcome variable of the study.
jcm-11-04638-t002_Table 2Table 2NOS quality score.StudiesSelectionComparabilityExposureTotal Quality Score
Author, YearRepresentativeness of the Exposed CohortSelection of the Non-Exposed CohortAscertainment of ExposureDemonstration That Outcome of Interest was not Present at Start of StudyComparability of Cohorts on the Basis of the Design or AnalysisAscertainment of Outcome Was Follow-Up Long Enough for Outcomes to OccurAdequacy of Follow Up of CohortsRecord of Physical Activity using PPAQAburezq 2020 [32] 111111107Antosiak-Cyrak 2019 [33] 001121117Chasan-Taber 2015 [34] 111121119Chasan-Taber 2014 [35] 111111118Davoud 2020 [36] 101120117Gebregziabher 2019 [37] 011121017Hailemariam 2020 [38] 110121017Harrold 2014 [39]101120117Ko 2016 [40] 010110115Lee 2016 [41] 101120106Lynch 2012 [42] 011120106Mourady 2013 [43] 111121119Okafor 2020 [44] 111120118Santos 2016 [45] 111111118Schmidt 2017 [46] 101111117Todorovic 2020 [47] 011121118Van der Waerden 2019 [48] 111121108Xiang 2019 [49] 010121117Yin 2019 [50] 011111117Zhang 2014 [51]011111117Record of Physical Activity using IPAQ Bertolotto 2010 [52]011110105Harizopoulou 2010 [53] 011121107Padmapriya 2015 [54] 011121118Rêgo 2017 [55] 111111118Román-Gálvez 2021 [56] 110120117Record of Physical Activity using KPAS Bacchi 2016 [57] 011120117Chasan-Taber 2007 [58]010121106Chasan-Taber 2008 [59]011121118Currie 2014 [60]011120117Fell 2009 [61] 110111117Fortner 2011 [62] 011110116Direct measurements of physical activity Di Fabio 2015 [63] 011111117Downs 2009 [64] 011111106Everson 2011 [65] 111121119Hawkins 2014 [66] 111120118Jiang 2012 [67] 111120107Morgan 2014 [68] 011121118Morkrid 2014 [69]111110117Rousham 2006 [70] 111111118Sinclair 2019 [71] 111120118Direct and indirect measurements with mixed studies of physical activity Chadonnet 2012 [72]011110116Cohen 2013 [73]011120117Kominiarek 2018 [74]011110105Medek 2016 [75]011110116
jcm-11-04638-t003_Table 3Table 3Record of Physical Activity using the PPAQ questionnaire during pregnancy.
Author, Year, CountryType of StudyNSampleGEST. AGEINST.PA REG.MainConclusionOther Variables[32]Aburezq2020KuwaitCross-sectional study653Pregnant women from Kuwait3T: >32 wkPPAQMET h /wkPA helps control weight, gestational blood pressure, and birth weight. Vigorous PA is more common in women without GDMSociodemographic, anthropometric, and pregnancy variables[33]Antosiak-Cyrak2019PolandObservational study267Polish pregnant women1T: 9 wk2T: 21 wk3T: 33 wkPPAQMET h/wkWomen prefer low to moderate intensity exercises.Women with previous children perform more PA.Sociodemographic and pregnancy variables[34]Chasan-Taber 2014USAProspective cohort study1276Hispanic pregnant women1T: 12 wk2T: 21 wk3T: 30 wkPPAQMET h/wkWomen who meet PA guidelines have a lower and controlled GWGSociodemographic, behavioral, psychosocial and pregnancy variables[35]Chasan-Taber 2015USAProspective cohort study1240Hispanic pregnant women12.4 wkPPAQMET h/wkPA performed before and early in pregnancy does not significantly reduce the risk of GHSociodemographic, behavioral, psychosocial and pregnancy variables[36]Davoud2020IranCross-sectional study256Iranian pregnant women1T: <13 wk2T:14–27 wk3T: >28 wkPPAQMET ·min/dayPA decreases during 2T and 3T.PA is related to quality of lifeSociodemographic, psychosocial and pregnancy variables[37]Gebregziabher2019 EthiopiaCross-sectional study458Ethiopian pregnant womenNDPPAQMET h/semNon-primiparous women with a higher level of education perform more PASociodemographic and pregnancy variables[38]Hailemariam2020EthiopiaCross-sectional study299Ethiopian pregnant women1T: <13 wk2T:14–27 wk3T: >28 wkPPAQMET h/wkWomen with a lower academic level and who work at home are at greater risk of being sedentarySociodemographic and pregnancy variables.[39]Harrod2014USALongitudinal cohort study826Pregnant women17 wk27 wk1 day postpartumPPAQMET h/wkHigher PA and total energy expenditure in the last stage of pregnancy are related to lower neonatal adiposity.Sociodemographic, pregnancy and anthropometric variables of mother and newborn.[40]Ko2016ChinaProspective descriptive study150Chinese pregnant women29 wk40 wkPPAQMET h/wkPA is higher in 2T and 3T.Women with low levels of PA have a greater chance of caesarean deliverySociodemographic, behavioral and anthropometric variables[41]Lee2016TaiwanCross-sectional study581Taiwanese pregnant women1T: 14–16 wk2T: 24–28 wk3T: 30–32 wkPPAQMET h/dayHigher energy expenditure in sports activities during 3T.Sociodemographic and anthropometric variables[42]Lynch2012USAProspective cohort study1355Hispanic pregnant women1T: 14–16 wk2T: 24–28 wk3T: 30–32 wkPPAQMET h/wkThe predominant energy expenditure is that of domestic and care activities. Multiparous women who underwent pre-pregnancy PA are less likely to be inactive.Sociodemographic, psychosocial and behavioral variables[43]Mourady2017LebanonCross-sectional study141Lebanese pregnant women1T: 14–16 wk2T: 24–28 wk3T: 30–32 wkPPAQMET h/wkLight PA is positively correlated with psychological health and social relationships.Sociodemographic, psychosocial, anthropometric behavioral and pregnancy variables[44]Okafor2020South AfricaCross-sectional study1082South African pregnant womenNDPPAQMET h/dayYounger women perform less PA. Single and unemployed women are less active.Sociodemographic, behavioral anthropometric variables[45]Santos2016PortugalLongitudinal prospective study118Portuguese pregnant women1T: <12 wk2T: 12–28 wk3T: >28 wkPPAQMET h/wkAF decreases significantly to a greater extent in 1T.Sociodemographic and anthropometric variables[46]Schmidt 2017GermanyObservational study83German pregnant women1T: 14–16 wk2T: 24–28 wk3T: 30–32 wkPPAQMET h/wkPA decreases during pregnancy despite showing the availability of the necessary information for it.Sociodemographic, behavioral anthropometric and pregnancy variables[47]Todorovic2020SerbiaCross-sectional study162Serbian pregnant women12 wkPPAQMET min/wkOne third of women have insufficient PA in 3T. Lower PA is associated with a lower educational level.Sociodemographic, behavioral and anthropometric variables[48]Van der Waerden2019France2 cohort studiesELFEEDEN5751745French pregnant women3T<24–28 wkPPAQMET h/wkLess PA and sedentary lifestyle seems to be associated with postpartum depression.Sociodemographic, psychosocial and pregnancy variables.[49]Xiang2019ChinaCross-sectional study1077Chinese pregnant women1T: <13 wk2T: 13–28 wk3T: >28 wkPPAQMET h/wkA high level of PA predominates, but not PE. Unemployed women without PA habits are more likely to fail to comply with PA guidelines, especially in 3T.Sociodemographic, behavioral, anthropometric and pregnancy variables.[50]Yin2019ChinaCross-sectional study1179Chinese pregnant women1T: <12 wk2T: 24–28 k3T: >32 wkPPAQMET h/wkDuring pregnancy, an inactive lifestyle predominates, with low intensity unit exercises.Sociodemographic, behavioral and anthropometric variables[51]Zhang2014ChinaCross-sectional study1056Chinese pregnant women1T: <13 wk2T: 14–27 wk3T: >28 wkPPAQMET h/dayWomen with a lower pre-pregnancy BMI, higher educational level and active husbands are more likely to perform PA and PE during pregnancy.Sociodemographic, anthropometric and pregnancy variables**N.:** sample size. **GEST. AGE.:** gestational age. **INST.:** instrument for measuring physical activity. **PA REG.:** record of physical activity. **T.:** trimester. **WK.:** week. **H.:** hour. **MET.:** metabolic equivalent of task. **MIN.:** minutes. **PA.:** physical activity. **GDM.:** gestational diabetes mellitus. **GWG.:** gestational weigh gain. **GH.:** gestational hypertension. **PE.:** physical exercise. **BMI.:** body mass index.
jcm-11-04638-t004_Table 4Table 4Record of Physical Activity using the IPAQ questionnaire during pregnancy.
Author, Year, CountryType of StudyNSampleGEST. AGEINST.PA REG.MainConclusionOther Variables[52]Bertolotto2010 ItalyObservational study268Italian pregnant women27 + 6 wkIPAQMET min/wkPA before pregnancy can lower the risk of GDM.Sociodemographic, behavioral, anthropometric and pregnancy variables[53]Harizopoulou2010 GreeceCross-sectional study160Pregnant women12 wkIPAQMET min/wkPhysical inactivity before and during early pregnancy is associated with an increased risk of developing GDMSociodemographic, anthropometric, pregnancy, childbirth and newborn variables.[54]Padmapriya2015SingaporeCohort study1247Chinese, Malarian and Indian pregnant women26–28 wkIPAQMET min/wkThe time spent on light, moderate and vigorous PA was reduced during pregnancy.Sociodemographic, psychosocial, anthropometric and pregnancy variables[55]Rêgo 2017BrazilCohort study1447Nulliparous pregnant women25–26 wkIPAQMET min/wkNo association was found between the level of PA in the 2T and 3T with adverse perinatal outcomes.Sociodemographic, pregnancy and delivery variables[56]Román-Gálvez2021SpainProspective cohort study463Healthy pregnant women1T: <14 wk2T: 24 wk3T: >32 wkIPAQMET min/wkTwo-thirds of women achieve enough PA. Energy expenditure decreases throughout pregnancy.Sociodemographic, behavioral anthropometric and pregnancy variables.**N.:** sample size. **GEST. AGE.:** gestational age. **INST.:** instrument for measuring physical activity. **PA REG.:** record of physical activity. **T.:** trimester. **WK.:** week. **H.:** hour. **MET.:** metabolic equivalent of task. **MIN.:** minutes. **PA.:** physical activity. **GDM.:** gestational diabetes mellitus. **GWG.:** gestational weigh gain. **GH.:** gestational hypertension. **PE.:** physical exercise. **BMI.:** body mass index.
jcm-11-04638-t005_Table 5Table 5Record of Physical Activity using the KPAS questionnaire during pregnancy.
Author, Year, CountryType of StudyNSampleGEST. AGEINST.PA REG.Main ConclusionOther Variables[57]Bacchi2016ItalyLongitudinal descriptive study236Caucasian pregnant women1T: 14–16 wk2T: 24–28 wk3T: 30–32 wkModified Kaiser QuestionnaireKPASActivity frequencyPA is generally low. In women with normal weight it increases in the 2T and 3T but in overweight women it remains stable.Sociodemographic, behavioral anthropometric and pregnancy variables.[58]Chasan-Taber 2007USAProspective study1231Latina pregnant women15 wk28 wkModified Kaiser QuestionnaireKPASActivity frequencyOccupational PA is higher in women with a high academic level and higher income. Domestic PA is higher in multiparous and older women.Sociodemographic, behavioral, anthropometric variables[59]Chasan-Taber2008USAProspective cohort study1006Hispanic pregnant women>24 wkModified Kaiser QuestionnaireKPASActivity frequencyA significant reduction in the risk of GDM is found in women who undergo some type of PA.Sociodemographic, behavioral, anthropometric and pregnancy variables[60]Currie2014CanadaProspective cohort study1749Canadian pregnant women20 wkModified Kaiser QuestionnaireKPASActivity frequencyPA together with an active lifestyle is associated with a lower appearance of fetal macrosomia.Sociodemographic, behavioral anthropometric variables, pregnancy and newborn.[61]Fell2009CanadaProspective cohort study1737Canadian pregnant women21+4 wkModified Kaiser QuestionnaireKPASActivity frequencyPA during the first 20 weeks of gestation is lower than pre-pregnancy PA.Sociodemographic, behavioral, anthropometric variables.[62]Fortner2011USAProspective cohort study1043Puerto Rican pregnant women15+5 wkModified Kaiser QuestionnaireKPASActivity frequencyRecreational PA in early pregnancy reduces the risk of GH.Sociodemographic, anthropometric, psychosocial and pregnancy variables.**N.:** sample size. **GEST. AGE.:** gestational age. **INST.:** instrument for measuring physical activity. **PA REG.:** record of physical activity. **T.:** trimester. **WK.:** week. **H.:** hour. **MET.:** metabolic equivalent of task. **MIN.:** minutes. **PA.:** physical activity. **GDM.:** gestational diabetes mellitus. **GWG.:** gestational weigh gain. **GH.:** gestational hypertension. **PE.:** physical exercise. **BMI.:** body mass index.
jcm-11-04638-t006_Table 6Table 6Direct measurements of physical activity during pregnancy.
Author, Year, CountryType of StudyNSampleGEST. AGEINST.PA REG.Main ConclusionOther Variables[63]Di Fabio2015USALongitudinal prospective study46American pregnant women18 wk35 wkAccelerometer Min/day in activity intensityTotal PA is higher in women who met recommendations for pre-pregnancy PA.Sociodemographic and anthropometric variables[64]Downs2009CanadaCohort study80Pregnant women20 wk32 wkPedometerSteps/dayWomen’s PA behaviors decreased from the second to third trimesters of pregnancySociodemographic and anthropometric variables[65]Everson2011USACross-sectional study359American pregnant women20–22 wkAccelerometer Min/day in activity intensityModerate/vigorous PA is higher in 1T and 2T compared to 3T.Sociodemographic, psychosocial and pregnancy variables.[66]Hawkins2014USACross-sectional study294Healthy pregnant women1T: <14 wk2T: 24 wk3T: >32 wkAccelerometer Min/day in activity intensityLight PA has a protective effect on CRP in 2T.Anthropometric and pregnancy sociodemographic variables.[67]Jiang2012ChinaCohort study919Healthy pregnant women18–28 wk29–35 wkPedometerSteps/day50% of women in 2T and 60% in 3T had low levels of PA.Sociodemographic and anthropometric variables[68]Morgan2014United KingdomProspective cohort study466Healthy pregnant womenNDAccelerometerMin/day in activity intensityA reduced PA is associated with instrumental deliveries. Being overweight and obese is related to pregnancy and childbirth problems.Sociodemographic, anthropometric, pregnancy and delivery variables.[69]Morkrid 2014 NorwayCohort study759Healthy pregnant women15 wkAccelerometerMin/day in activity intensityHigher level of PA in early pregnancy reduces the risk of GDM development.Sociodemographic, anthropometric and pregnancy variables.[70]Rousham 2006United KingdomProspective cohort study57Pregnant womenwk:1216253438AccelerometerMin/day in activity intensityPA levels are reduced during pregnancy.Sociodemographic variables[71]Sinclair 2019CanadaCohort study70Canadian pregnant women1T: 16–18 wk 2T: 24–26 wk3T: 32–34 kAccelerometer Min/day in activity intensityA higher level of sedentary lifestyle is associated with a higher level of perceived stress.Sociodemographic variables**N.:** sample size. **GEST. AGE.:** gestational age. **INST.:** instrument for measuring physical activity. **PA REG.:** record of physical activity. **T.:** trimester. **WK.:** week. **H.:** hour. **MET.:** metabolic equivalent of task. **MIN.:** minutes. **PA.:** physical activity. **GDM.:** gestational diabetes mellitus. **GWG.:** gestational weigh gain. **GH.:** gestational hypertension. **PE.:** physical exercise. **BMI.:** body mass index.
jcm-11-04638-t007_Table 7Table 7Direct and indirect measurements with mixed studies of physical activity during pregnancy.
Author, Year, CountryType of StudyNSampleGEST. AGEINST.PA REG.MainConclusionOther Variables[72]Chandonnet2012 CanadaCross-sectional study49Canadian pregnant women with obesity
1T: 13 wk2T: 25 wk3T: 35 wk
PPAQAccelerometerMET h/wkMin/day in activity intensityPA is reduced during pregnancy. The highest energy expenditure occurs in housework and sedentary activities.Sociodemographic, behavioral and anthropometric variables.[73]Cohen 2013CanadaObservational study54Canadian pregnant women26 wkPPAQPedometerMET h/daySteps/dayWomen with a goal to perform PA are more likely to meet the guidelines.Sociodemographic, anthropometric and pregnancy variables.[74]Kominiarek 2018USAObservational study49Hispanic and american pregnant women28 wk36 wkPPAQAccelerometerMET h/wkMET min/daySteps/dayPA is reduced and sedentary activity increases as the pregnancy progresses.Sociodemographic, behavioral and anthropometric variables.[75]Medek 2016 IcelandObservational study217Icelandic pregnant women24- 28 wkIPAQPedodometerMET min/wkSteps/dayVigorous PA appears to be beneficial to maternal glucose tolerance, both in BMI and overweight and obese women.Sociodemographic, behavioral and pregnancy variables.**N.:** sample size. **GEST. AGE.:** gestational age. **INST.:** instrument for measuring physical activity. **PA REG.:** record of physical activity. **T.:** trimester. **WK.:** week. **H.:** hour. **MET.:** metabolic equivalent of task. **MIN.:** minutes. **PA.:** physical activity. **GDM.:** gestational diabetes mellitus. **GWG.:** gestational weigh gain. **GH.:** gestational hypertension. **PE.:** physical exercise. **BMI.:** body mass index.


## 3. Results

The flow chart shows the results obtained in relation to each of the study development phases: identification, exploration, eligibility, and inclusion. Of a total of 1451 studies that were identified, 836 were screened, and 110 of them were full text assessed for eligibility. A total of 44 studies [32,33,34,35,36,37,38,39,40,41,42,43,44,45,46,47,48,49,50,51,52,53,54,55,56,57,58,59,60,61,62,63,64,65,66,67,68,69,70,71,72,73,74,75] were finally analyzed and included for synthesis.

### 3.1. Quality Assessments

The quality assessments of the included studies are summarized in Table 2. The median quality score of the studies was 7.12 (range 5 to 9) after removing low-quality studies. With this assessment, 22.73% of the studies were rated as moderate quality (n = 10) [40,41,42,52,58,62,64,72,74,75], and 77.27% were rated as high quality (n = 34) [32,33,34,35,36,37,38,39,43,44,45,46,47,48,49,50,51,53,54,55,56,57,59,60,61,63,65,66,67,68,69,70,71,73].

In detail, when analyzing the selection quality factors, 52.27% of the studies [33,37,40,42,47,49,50,51,52,53,54,57,58,59,60,62,63,64,68,72,73,74,75] had flaws in the representativeness of the exposed cohort due to the calculation of the sample size being unusual. In terms of comparability, all studies displayed at least one control factor for the comparability of the cohorts based on the design or analysis. Finally, in relation to the evaluation of the quality of the results, 40.90% of the studies [36,39,40,41,42,44,52,56,57,60,62,66,67,69,71,72,73,74,75] lacked evidence in the control of the factor or ascertainment of the outcome.

### 3.2. Characteristics of the Included Studies

In a global perspective, 44.44% of the studies (n = 12) [33,37,38,42,44,47,49,51,56,57,58,64] reported, as a primary result, that PA levels were directly and positively related to maternal educational level, and the remaining 55.56% [32,34,35,36,39,40,41,43,45,46,48,50,52,53,54,55,59,60,61,62,63,65,66,67,68,69,70,71,72,73,74,75] showed, on one hand, the relationship of PA levels with other maternal factors; on the other hand, they focused on describing the practice of PA during pregnancy. PA carried out during this period is recorded as mostly in domestic activities [41,42,58,72].

In general, the analyzed studies covered two types of studies with different themes: on one hand, 61.36% (n = 27) [33,34,36,37,38,41,42,44,45,46,47,49,50,51,54,56,57,58,61,63,64,65,67,70,72,73,74] showed the descriptors of patterns and habits of physical activity during pregnancy, and on the other hand, 38.64% (n = 17) [32,35,39,40,43,48,52,53,55,59,60,62,66,68,69,71,75] showed the effect of a physically active lifestyle during the nine months of gestation on different maternal-fetal parameters. Over half of the articles (59.09%) (n = 26) [33,34,36,37,38,41,42,44,45,46,47,49,50,51,54,57,58,61,63,64,65,67,70,72,73,74] found levels below what is indicated in the international recommendations, namely 150 min per week of moderate PA.

Only one article found that two-thirds of the sample met the recommendations [56]. The remaining 17 articles also found inadequate PA levels associated with pregnancy problems (gestational hypertension, diabetes, depression, anxiety, or fetal macrosomia). Low to moderate intensity of gestational PA was registered more frequently than the vigorous one [33,43,50,66], without the weekly exercise being sufficient to meet the international standards.

The gestational period in which the information was collected in the studies is diverse, as is the methodology for describing it. Articles that compared physical activity during the three trimesters of pregnancy stand out [33,36,38,39,41,42,43,45,46,49,50,51,56,57,66,70,71,72], indicating the gestational weeks in which the data were collected. The other group of articles [32,33,34,35,40,43,44,45,58,59,60,61,62,63,64,65,66,67,69,73,74,75] collected data on PA performed in the first weeks of pregnancy (>15 weeks) middle of pregnancy (20–24 weeks) or at the end of pregnancy (>26 weeks) in a specific way. Physical activity decreases throughout pregnancy compared to pre-gestational PA [46,47,50,54]. Specifically, low and inadequate PA was found during the first trimester [60,61,70] and there was a progressive decrease from the second to the third trimester [36,49,56,64,65,67,72,74]. In contrast, 2 articles found a higher PA, without being enough, during the last trimester of pregnancy [40,45].

The included studies were found in a total of 22 countries and relate to a large sample of pregnant women (n = 28,728). Specifically, the largest number of studies have been carried out in Asia (10) [32,36,40,41,43,49,50,51,54,67], North Europe (n = 7) [33,46,48,68,69,70,75], South Europe (n = 6) [45,47,52,53,56,57], North America (n = 17) [34,35,39,42,58,59,60,61,62,63,64,65,66,71,72,73,74], Africa (n = 3) [37,38,44] and South America (n = 1) [55].

Asian countries find lower levels of PA during pregnancy and a progressive decrease in it, with the exception of the study by Ko (2016) [40]. The physical measurement study stands out, finding that 50% during the 2nd trimester and 60% during the third trimester do not reach the suggested recommendations.

As for the European continent, a decrease in the PA level during pregnancy is reflected both for the countries of North and South Europe. A lower PA is associated with mental [48] and physical [68] problems, as well as evidence for both the northern region [69] and the southern region [52,53] that PA from early pregnancy prevents complications, such as gestational diabetes. Specifically, it is recorded in South Europe that two thirds of the pregnant population do not reach the optimal energy expenditure during pregnancy [56].

In studies conducted in the North American area, PA generally decreases and sedentary lifestyle increases during pregnancy; however, a difference is visible between behavioral self-report studies that find lower PA in the first trimester [61], and studies with direct physical measurements [64,65], which record lower levels as gestation progresses. Similarly, it is found that performing PA only at the beginning of pregnancy lacks benefits [35]. Nevertheless, performing PA from the beginning and continuing throughout the pregnancy prevents complications such as gestational diabetes [34,59] or gestational hypertension. [62], as in European countries.

Studies conducted in Africa mainly associate the practice of PA with educational level and multiparity, with educated women and those with previous children being the most active [37,38,44]. Finally, the only study found in South America [55] shows no association between PA and perineal outcomes.

The study designs that are presented varied in their nomenclature, with 43.18% of the studies (n = 19) defined as cohort studies [34,35,39,42,48,54,55,56,58,59,60,61,62,64,67,68,69,70,71] and 34.09% defined as cross-sectional studies (n = 15) [32,36,37,38,40,43,44,47,49,50,51,53,65,66,72]. The remaining studies have been classified by their authors as observational [33,46,52,73,74,75] and prospective longitudinal [45,57,62]. Cohort studies base the selection of the cohort on the specific population of the corresponding regions.

Related to pregnancy outcomes, maternal variables that were collected were divided into sociodemographic (age, academic level, purchasing power, job occupation, marital status, parity or previous miscarriages) anthropometric (prepregnancy BMI, prenatal weight gain, height, waist, hip circumference, maternal BMI at delivery), psychosocial and behavioral (sleep habits, food intake, smoking or alcoholism), pregnancy (parity, ethnicity, age, education level, marriage, smoking previously and during pregnancy, employment status, household income, weight gain, skinfolds, body circumferences, chronic disease history, gestational diabetes, blood pressure, gestational hypertension, preeclampsia), perinatal (gestational age at delivery, birth <37 weeks, type of delivery, duration or injuries) and newborn (perinatal mortality, birth weight (<2500g, 2500–4000g, >4000g), body fat mass, Apgar 1, Apgar 5, length or head circumference) data. Studies found that performing PA helps control gestational weight gain [32], reduced risk of gestational diabetes [34,52,53,59,69] and gestational hypertension [62], lessened c-sections [40] and instrumental delivery [68], lower neonatal adiposity [39] and macrosomia [60] and stress [71] during pregnancy and lessened postpartum depression [48].

In addition, it is relevant to show that 13 studies [34,35,36,42,43,44,48,51,56,62,67,69,71] administered other questionnaires on nutrition, sleep or depression to address and fully explore the complexity of the multifactorial course of pregnancy.

Research studies were divided into three different groups: studies with indirect methods which use validated questionnaires for data recording, studies with direct methods using on-site technological material to take measurements and mixed studies with direct and indirect methods of physical activity quantification. The data collection has been similar in all the studies analysed, since it is given by the scientifically validated protocol of each instrument used.

#### 3.2.1. Studies with Indirect Methods of Physical Activity Quantification

Regarding the data collection instruments, within the indirect measurements, the use of the “Pregnancy Physical Activity Questionnaire” (PPAQ) was predominant, as a validated and reference questionnaire among the pregnant population (Table 3), followed by the “International Physical Activity Questionnaire” IPAQ (Table 4) and the “Modified Kaiser Questionnaire” KPAS (Table 5). These questionnaires are reliable and validated in pregnant women and provide information on the time spent on activities in the life of pregnant women, including items on family, work, or sports/exercise.

The main difference found in the collection of information resides in the fact that the PPAQ records of PA use METS hours/week, the IPAQ uses METS minutes/week and the KPAS has its own measurement on a Likert-type scale, making it difficult to compare the quantitative results among them. Furthermore, among the studies that used the PPAQ as a measurement instrument, 5 collected PA in METS hours/day, making extrapolation of the results even more difficult.

#### 3.2.2. Studies with Direct Methods of Physical Activity Quantification

The main instruments for the objective measurement of PA during pregnancy that are reflected in the scientific literature are the use of accelerometers and pedometers, with 7 and 2 articles in each case, respectively (Table 6). The measurement system for the accelerometer is the min/day that are invested for each activity and for the pedometers, the amount of steps/day. In addition, a wide heterogeneity was observed in the duration of data collection, ranging between 3 and 7 consecutive days of recording the activity with these devices in the studies.

#### 3.2.3. Mixed Studies with Direct and Indirect Methods for Quantifying Physical Activity

Publications that used both (4 studies) both direct and indirect measures focused on a more complete view of PA patterns and combined the two reference questionnaires (PPAQ and IPAQ) with objective data from accelerometers and pedometers (Table 7). The data collection system was identical to what was stated in the previous sections.

## 4. Discussion

This study aimed to examine the level of PA in pregnant populations from different regions of the world. When attempting a global analysis of the different studies, one of the main questions from a scientific point of view is the diversity found in the way of measuring the gestational PA as seen in the included tables. It is interesting to note that many of these studies, in addition to the assessment of the PA, combined the analysis of certain maternal-fetal parameters, which provides us with an additional analysis to consider. Regarding the division by geographical regions, despite finding differences in the study modalities and associations between registered variables, possibly due to the type of research that has been performed and the economic resources to carry it out, low patterns of PA are widely found.

In this sense, it can be observed that the studies that use direct measures of physical activity using technological elements are carried out in high-income countries, mainly in North America [63,64,65,66,71] and North Europe [68,69,70]. However, the compatibility of the results can become complex due to variations in the methodology, registration, and data processing [76] so the conclusions should be interpreted with caution. In contrast, studies conducted in the African [37,38,44] or South American [55] regions using self-report questionnaires associate this physical activity with maternal educational levels and parity, showing a great intrinsic social problem in these geographical areas. However, the quantity and quality of studies in low-income countries is scarce, so the use of a specific methodology to study the prevalence of PA and the increase in socioeconomic resources to obtain data that can be extrapolated worldwide is vital.

Similarly, previous studies with general populations, show a deep variation in trends in inactivity across regions, income groups, and countries [77]. Higher rates of insufficient physical activity have been recorded in high-income countries while East and Southeast Asia maintain better physical levels [77]. In contrast, women and populations with limited economic resources tend to be less physically active than those with greater resources [78] so this trend could be similar with the pregnant population.

In addition to the cultural paradigm associated with each geographical region, it is worrying that approximately 40% of the female population meets the minimum recommendations for weekly PA [79], evidencing as a latent problem among the population of reproductive age. Added to this is that, currently during the COVID-19 pandemic, the lowest PA levels are more frequent in women, being a potential risk group for physical inactivity [80] during all stages of life, especially during pregnancy.

It is clearly essential to educate women that physical activity promotes improved cardiovascular function, decreased risk of gestational diabetes mellitus, hypertension, and a lower percentage of body fat in the mother, increased gestational age, and improved neurodevelopment in the child [81]. Thus, a recommendation for PA with early onset and throughout pregnancy may be key to improving maternal-fetal parameters [34,35,59,62].

From a methodological point of view, the studies reviewed have great variation as to when they evaluated PA during the pregnancy which makes it difficult to compare across studies and represents an important future research topic. This problem was exposed in the study by Chan (2019) on regulated PA programs where inconsistencies in findings hamper the drawing of firm conclusions [82]. Therefore, an important scientific challenge exists in this area, needing future research.

As stated, the analysis of each of the aforementioned tables offers us different ways of studying and recording gestational PA, which at the same time causes significant confusion and makes data generalization impossible. However, in each of the cases, it can be verified that pregnancy is a period with a certain decrease, in some cases, the total absence, of a regular practice of physical activity. Although the recommendations were not followed, the realization of the minimum level of physical activity during pregnancy is supported by most of the articles reviewed and is in agreement with other previous studies [83,84].

At a general level, the results obtained show that, in most populations studied, PA levels do not reach universal basic recommendations to achieve the already proven benefits of physical practice in maternal, fetal, and newborn outcomes. This has been previously exposed by Borondulin and colleagues (2008) who showed that only a small proportion of pregnant women reached the recommended level of PA [85], and by Lindqvist and colleagues (2016) who found that less than half of the sample met the guidelines [86]. Likewise, scientific studies with the adult population find that less than 10% of adults perform 150 min per week of moderate PA [87]. Specifically in pregnant women, it has been observed that, maintaining levels of light physical activity can promote wellbeing [88] as well as controlling the presence of gestational diabetes, even when starting from a lower intensity physical activity measured with accelerometry [89]. However, previous studies with direct measurements of quality accelerometry in this population are limited and the studies analyzed do not provide generalized conclusions.

In the same way, it was observed that higher levels of physical practice are associated with higher socioeconomic and cultural conditions, and many of the studies that were analyzed remind us of the positive effects of moderate physical practice on maternal parameters, not only of a physical or physiological nature but also psychological and emotional. This has been demonstrated by studies on this subject [90,91,92]. Together, these results show the need to establish a minimum guideline of moderate PA during pregnancy as a basic and fundamental element for the health of future populations.

Even when a more detailed analysis was carried out and attention was focused on the European population, studies developed in Germany [46], Spain [56], France [48], Greece [53], Italy [52,57], Iceland [75], Norway [69], Portugal [45], the United Kingdom [68,69,70], Serbia [47] and Poland [33] clearly and specifically show this decrease in PA in the gestational period. Some of these studies also reaffirmed the beneficial effects mentioned previously that are associated with a more active pregnancy.

Beyond the European analysis, in our opinion, the excellent study carried out by Evenson and Wen in 2011 [65] with 359 pregnant American women aged ≥16 years, who wore an accelerometer for 1 week, is remarkable. This study described the prevalence and correlations between physical activity and sedentary behaviors measured among these pregnant women. For this, in addition to the accelerometer, cross-sectional data collected from the United States National Health and Nutrition Survey (NHANES) from 2003 to 2006 were used. The authors found that most monitored pregnant women spent more than half of the day performing sedentary behaviors and did not comply with the established physical activity recommendations. This sedentary behavior was directly associated with lower levels of economic income and was less significant at high levels of income.

The central idea of a physically active pregnancy as a health promoter continues to be a necessary objective and directly related to universal strategies such as the Sustainable Development Goals proposed by the United Nations (SDG) [93], specifically with objectives 3 (to ensure healthy lives and promote well-being for all at all ages) and 5 (to achieve gender equality and empower all women and girls); but this undoubtedly still needs greater promotion, especially from institutions in charge of the health of pregnant women. The lack of progress in the different regions could explained by the fact that, although more than 70% of countries have an operational physical activity policy, the scale and scope of its implementation has not yet had a national impact [94,95].

Future studies in this scientific field should focus on the prevalence of physical activity during pregnancy in different populations and its relationship with different maternal, fetal and newborn parameters during and after the gestational period. In addition, it is necessary to pay attention to determining the barriers that pregnant women face in their limited access (currently) to the regular and safe practice of physical activity and even more if this low level of practice is in some way affecting the health of the mother and child. The main limitation observed in the present work is the diversity of intervention designs presented by the analyzed studies, which logically prevents generalization in the analysis and discussion of the results, affecting, to a certain extent, the validity of the conclusions reached. Another limitation to consider is related to the different gestational periods studied by the different authors, which naturally causes significant heterogeneity in the targeting of the periods of pregnancy in which the results of each investigation were obtained.

This article provides an updated view of worldwide data on the level of physical exercise performed by different pregnant populations around the world. Health practitioners should continue to promote moderate physical activity during pregnancy; in fact, not reaching 150 min of weekly moderate physical activity could mean significant risks in all areas of maternal health (physiological, mental, emotional) and child health. From the research point of view, new clinical guidelines, and recommendations for exercise during pregnancy, should be developed in different settings.

## 5. Conclusions

The results of the present study allow us to conclude that even though in the last 15 years there has been a significant increase in physical practice in the pregnant population, the current levels observed in the present review are still very far from the universal recommendations proposed by international organizations in their related publications.

## Figures and Tables

**Figure 1 jcm-11-04638-f001:**
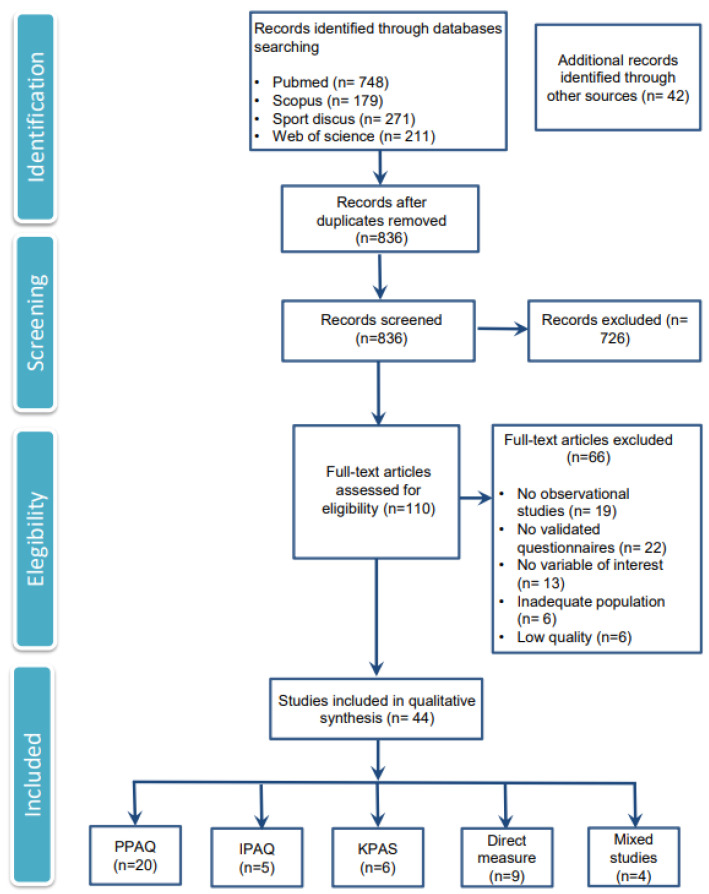
Flow chart of the results in each of the study development phases.

**Table 1 jcm-11-04638-t001:** PICOS framework.

PICOS	DEFINITIONS
**Population**	Healthy pregnant women.
**Interventions**	PA assessment carried out during pregnancy.
**Comparison**	Baseline data based on not practicing physical activity.
**Outcome**	The main variable of the study should record the PA developed during pregnancy.
**Study design**	Observational studies.

## Data Availability

Not applicable.

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
