# Peer review of "Level of Physical Activity in Pregnant Populations from Different Geographic Regions: A Systematic Review"

_jcm, 2022, doi:10.3390/jcm11154638_

Round 1

Reviewer 1 Report

Manuscript-JCM-1784687

Level of physical activity in pregnant populations from different geographic regions: a systematic review.

The manuscript describes “the objective to examine the level of physical activity (PA) of pregnant populations across the world.” Line 66. This implies both a qualitative analysis of the research design and a summary of the quantitative information.  While this is a worthwhile endeavor the manuscript has weaknesses in both areas.

Qualitative analysis:

1.      The regions are not grouped well. The culture of physical activity is different by region: regular walking during summer in a middle eastern dessert or a wintery environment of Scandinavia is very different. Give more value to the title and subgroup the studies to analogous climates.

2.      Poor quality score threshold is not stated (< 7?).

3.      The articles were screened by two authors (line 135) and consensus was required for inclusion, A third arbiter who was blinded to the previous determinations should be used for a tie breaker.

4.      The tables reflect discrepancies in “Type of Study” and Gestational age. For example, studies by Chasen-Tabor (2014 #32, 2015# 33)  are described as prospective cohorts with one point in time evaluation (12 weeks GA). Please explain. There are other discrepancies.

5.      The studies should be grouped by design quality and compared: prospective cohort with physical measurement as well as behavior self-report, prospective cohort with behavioral self- report only, cross sectional, observational. The cross sectional designed studies should stratify the results by at least trimester.

6.      Please explain why the results of control groups in intervention studies are not included.

7.      The ACOG committee opinion is from 2020 (ref 14). Many of the reference occurred before this reference resource was published. Why was this standard used? Did the individual countries have published standard?

Quantitative:

1.      Basic measurements of physical outcomes of mothers and babies is missing for the studies: these measurements include maternal BMI at delivery, prenatal weight gain, GA at delivery , birth < 37wks, BW < 2500grams, Apgar at 5 minutes < 7,  perinatal mortality, birthweight > 4000 gm.

2.      None of the data collected in lines 214-219 are reported.

3.      The lack of adherence to activity guidelines is not correlated to GA at measurement, despite numerous references to decreased activity across pregnancy.

General comments:

1.      Please review the document for spelling (pedometer vs podometer- Tables).

2.      The discussion should report PA for reproductive aged, non-pregnant women. Maybe the problem is the female (and male) population, are not consistent participation in PA.

Author Response

Many thanks to the reviewers for their time and effort in correcting this manuscript. Below, the questions are answered with the changes made and the explanation of these.

In the same way, the certificate of edition in English made by AJE is inserted at the end of the document.

Qualitative analysis:

  1. The regions are not grouped well. The culture of physical activity is different by region: regular walking during summer in a middle eastern dessert or a wintery environment of Scandinavia is very different. Give more value to the title and subgroup the studies to analogous climates.

In response to the reviewer's comment, the results and discussion section has been restructured, dividing by geographic region. (Lines 190-255)

  1. Poor quality score threshold is not stated (< 7?).

Lines 138-146 explain how the studies have been selected following the scale. In this sense, studies with moderate and high quality have been maintained and those with lower scores have been discarded. To assess this, the criteria for applying the scale were followed and, in addition to the total value of study quality, the partial scores of the items in each of the three categorized groups were taken into account.

  1. The articles were screened by two authors (line 135) and consensus was required for inclusion, A third arbiter who was blinded to the previous determinations should be used for a tie breaker.

This step of the data extraction was considered but not written. Now, we have added the corresponding information (lines 141-142)

  1. The tables reflect discrepancies in “Type of Study” and Gestational age. For example, studies by Chasen-Tabor (2014 #32, 2015# 33)  are described as prospective cohorts with one point in time evaluation (12 weeks GA). Please explain. There are other discrepancies.

According to the text of both articles, they have selected as a cohort, in both cases based on Hispanic women, so gestational age has not been the criteria for said study. Regarding the physical activity records, the time of evaluation and data collection has been shown in the table extracted directly from the text of the revised articles.

  1. The studies should be grouped by design quality and compared: prospective cohort with physical measurement as well as behavior self-report, prospective cohort with behavioral self- report only, cross sectional, observational. The cross sectional designed studies should stratify the results by at least trimester.

When restructuring by geographic area and organizing the results based on it, we consider the main organization based on that criterion instead of based on the type of study. In addition, the studies have been specified in more detail and a view of the cohorts has been provided, which is commented on lines 250-255

  1. Please explain why the results of control groups in intervention studies are not included.

            In this sense, the studies whose design is experimental (RCT or non-RCT) with control groups have not been included in the review, for which reason data of this nature are not available in the selected studies.

  1. The ACOG committee opinion is from 2020 (ref 14). Many of the reference occurred before this reference resource was published. Why was this standard used? Did the individual countries have published standard?

References from 2019, 2020 and 2021 have been used to find the most up-to-date point of view. Nevertheless, these recommendations follow the lines of other previous clinical guidelines. Following your recommendation, a paragraph was added to see the similarity with the previous context, providing information from the 2000 and 2002 ACOG clinical guidelines, just prior to the first articles of the systematic review (these articles are between 2006 and 2021) which are a worldwide reference. This information was added in lines 59-63.

Quantitative:

  1. Basic measurements of physical outcomes of mothers and babies is missing for the studies: these measurements include maternal BMI at delivery, prenatal weight gain, GA at delivery, birth < 37wks, BW < 2500grams, Apgar at 5 minutes < 7,  perinatal mortality, birthweight > 4000 gm.

Following the reviewer's comment, the information has been added to the wording in the results. (Lines 256-265)

  1. None of the data collected in lines 214-219 are reported.

Following the reviewer's comment, this secondary information has been added to the wording in the results. (Lines 267-271)

  1. The lack of adherence to activity guidelines is not correlated to GA at measurement, despite numerous references to decreased activity across pregnancy.

The relative information about physical activity and its decrease appears between lines 211-221.

General comments:

  1. Please review the document for spelling (pedometer vs podometer- Tables).

Following the comment, it has been revised throughout the document and has been modified by the corresponding word in the tables.

  1. The discussion should report PA for reproductive aged, non-pregnant women. Maybe the problem is the female (and male) population, are not consistent participation in PA.

Following this comment from the reviewer, that part has been modified and added. (Lines 387-392)

Reviewer 2 Report

I appreciate the efforts taken by the authors in conducting such an interesting systematic review. However, I have few concern

1.       Why it is mentioned in the title as population from different geographical regions. In the aim and in the discussion the same has been repeated. However, there is no comparison between various regions in the results. This should be added to the results and should be discussed in the discussion section (noticed a few lines in the discussion)

2.       Add PROSPERO registration details in the methodology

3.       Search criteria are not clear. Are the same criteria used for all database? Provide search criteria used for each database separately. I recommend adding a separate table for this

4.       Why NOS was used for risk bias assessment?

5.       Line 170,174,190 – write the number of articles in addition to the percentage. The comment is applicable to the entire manuscript

6.       The discussion has to be improved by discussing each finding in the results. Regional-wise variation also has to be discussed in the discussion section after its inclusion in the results.

Author Response

Many thanks to the reviewers for their time and effort in correcting this manuscript. Below, the questions are answered with the changes made and the explanation of these.

In the same way, the certificate of edition in English made by AJE is inserted at the end of the document.

  1. Why it is mentioned in the title as population from different geographical regions. In the aim and in the discussion the same has been repeated. However, there is no comparison between various regions in the results. This should be added to the results and should be discussed in the discussion section (noticed a few lines in the discussion)

In response to the reviewer's comment, the results and discussion section has been restructured, dividing by geographic region. (Lines 190-255) (Lines 366-392, Lines 451-453)

  1. Add PROSPERO registration details in the methodology

Following the reviewer's comment, the requested information has been added (lines 76-78)

  1. Search criteria are not clear. Are the same criteria used for all database? Provide search criteria used for each database separately. I recommend adding a separate table for this

Following the reviewer's comment, this section has been clarified in the corresponding part (lines 83-84). To carry out the search, a common criteria has been established based on the objective of the study.

  1. Why NOS was used for risk bias assessment?

We have used this tool because it is specific to assess the quality of non-randomized studies and it fits the typology of our systematic review, which does not include this type of study.

  1. Line 170,174,190 – write the number of articles in addition to the percentage. The comment is applicable to the entire manuscript

Following the reviewer's comment, corrections have been made to the entire manuscript.

  1. The discussion has to be improved by discussing each finding in the results. Regional-wise variation also has to be discussed in the discussion section after its inclusion in the results.

Following this comment from the reviewer, that part has been modified and added. (Lines 366-392, Lines 451-453)

Round 2

Reviewer 2 Report

The paper has been improved. Still there are few issues which is not addressed properly

The search criteria, still not clear .Please provide search criteria used for each data base. Search should be based on PICOS .It is highly recommended to keep a separate table or supplementary table for search criteria 

Author Response

Thank you very much for your review.

Following the comment, more information has been added on lines 83-85 and 140-141 detailing how the search process was carried out.

Similarly, the document with the search terms by databases has been added to the supplementary material.